# The Barriers and Facilitators to the Application of Non-Invasive Brain Stimulation for Injury Rehabilitation and Performance Enhancement: A Qualitative Study

**DOI:** 10.3390/neurosci6030072

**Published:** 2025-08-01

**Authors:** Chris Haydock, Amanda Timler, Casey Whife, Harrison Tyler, Myles C. Murphy

**Affiliations:** 1Nutrition and Health Innovation Research Institute, School of Medical and Health Sciences, Edith Cowan University, Joondalup, WA 6027, Australia; c.haydock@ecu.edu.au (C.H.);; 2Research Department, Child and Adolescent Health Service, Nedlands, WA 6009, Australia; amanda.timler@health.wa.gov.au; 3High Performance Department, West Coast Eagles Football Club, Lathlain, WA 6100, Australia; 4Independent Consumer, Joondalup, WA 6027, Australia; 5Institute for Health Research, The University of Notre Dame Australia, Fremantle, WA 6160, Australia

**Keywords:** tDCS, interview, clinical practice, validity, utility

## Abstract

Introduction: Despite clinical evidence for efficacy, there has been minimal uptake of transcranial direct current stimulation (tDCS) for musculoskeletal conditions. Thus, our objective was to explore the perceptions and experiences of people living with lower-limb musculoskeletal injury as well as healthy physically active populations and relate this to the usage of tDCS and key aspects of tDCS design that would improve the capacity for implementation. Methods: We conducted a qualitative descriptive study of 16 participants (44% women) using semi-structured focus groups to identify the descriptions and experiences of people living with lower-limb musculoskeletal injury and healthy physically active populations. A thematic template was used to create a coding structure. Codes were then grouped, and key themes were derived from the data. Results: Four primary themes were identified from focus groups. These were (i) the impact of musculoskeletal injuries on health and quality of life, (ii) performance and injury recovery as facilitators to using tDCS, (iii) barriers and facilitators to tCDS application and (iv) design and aesthetic factors for a tDCS device. Discussion: Our qualitative descriptive study identified four themes relevant to the successful implementation of tDCS into rehabilitative and performance practice. To increase the likelihood of successful tDCS implementation, these barriers should be addressed and facilitators promoted. This should include innovative approaches to device application and structure that allow for a stylish, user-friendly design.

## 1. Introduction

Many musculoskeletal conditions, such as anterior cruciate ligament rupture [1], osteoarthritis [2], or tendinopathy [3], result in substantial physical impairment. These physical impairments are directly related to pain, disability and poor quality of life [1,4,5]. The recommended management of these conditions, and their physical impairments, is predominantly based around implementing exercise rehabilitation [6,7,8,9,10,11,12,13,14].

However, in addition to physical impairments, there are also substantial neurophysiological impairments present in people with musculoskeletal conditions [15,16,17,18,19,20,21,22,23]. One of the most prominent musculoskeletal impairments is impaired voluntary activation [24,25], which reduces the body’s ability to produce strong muscle contractions and is linked to higher pain levels [26]. However, current rehabilitation programmes do not improve voluntary activation [13]. This is problematic when the primary aim of exercise rehabilitation is often to increase physical function via resistance training protocols that involve strong muscle contractions [8,11,27,28,29,30,31]. Another neurophysiological feature concerning people with chronic musculoskeletal conditions, such as osteoarthritis or ligament rupture, is excessive motor cortex inhibition [17,32,33,34,35]. Clinical studies have demonstrated a relationship between voluntary activation and the degree of motor cortex inhibition with lower voluntary activation being associated with higher levels of inhibition [36]. However, like voluntary activation, the evidence for current best practice (i.e., exercise rehabilitation) at improving cortical inhibition in people with musculoskeletal conditions is lacking.

It has been proposed that non-exercise interventions that reduce motor cortex inhibition might be an effective adjunct to improve health outcomes in people with chronic musculoskeletal conditions [37]. Specifically, non-invasive brain stimulation interventions have been proposed as an adjunct to exercise rehabilitation that may alter cortical activity [38,39,40,41,42,43,44,45]. Non-invasive brain stimulation has been shown to directly alter the amount of motor cortex inhibition [45,46], and the efficacy of non-invasive brain stimulation appears to be associated with the number of treatment sessions [47].

Clinical trials have demonstrated the benefits of non-invasive brain stimulation for musculoskeletal conditions, such as knee osteoarthritis and anterior cruciate ligament reconstruction [40,45,48,49,50]. One of the mechanisms by which non-invasive brain stimulation is proposed to work in people with chronic pain is restoring impaired pain modulatory mechanisms, such as conditioned pain modulation [48,51]. Despite the evidence for efficacy in reducing pain, there has been minimal uptake of non-invasive brain stimulation interventions, such as transcranial direct current stimulation (tDCS), for chronic musculoskeletal conditions in clinical practice [52].

The poor uptake of tDCS to date may reflect device design, in which current devices are better designed to stimulate the dorsolateral prefrontal cortex rather than the motor cortex. Alternatively, a lack of uptake may be due to the general hesitancy of the population to utilise brain stimulation technology for fear of adverse events. Thus, the primary objective of this study was to explore the perceptions and experiences of people living with lower-limb musculoskeletal injury as well as healthy physically active populations and relate this to the usage of tDCS and key aspects of tDCS design that would improve the capacity for implementation.

## 2. Methods

### 2.1. Design

We conducted a qualitative descriptive study using semi-structured focus groups to identify the descriptions and experiences of people living with lower-limb musculoskeletal injury and healthy physically active populations.

### 2.2. Reporting Criteria

This study is reported using the consolidated criteria for reporting qualitative studies (COREQ) [53].

### 2.3. Concepts of Interest

To address the objectives of our study, we developed a series of semi-structured interview questions that were presented to participants (Appendix A). This included a total of six questions, each question relating specifically to our concepts of interest. We had initially planned for homogenous focus groups of physically active versus injured populations. However, it became clear during recruitment that most high-level athletes had a substantial injury history, and this would be challenging. Thus, questions were adapted for individuals based on whether they had reported an injury during the opening section of the focus group. Thus, the homogenous nature of our focus groups changed to experiencing a substantial injury, regardless of the level of physical activity (general population or elite athlete).

### 2.4. Research Team and Reflexivity

#### 2.4.1. Personal Characteristics

One member of the research team (MCM) with substantial experience in qualitative research [54,55,56], conducted all of the focus groups, with his personal characteristics described in Table 1. The facilitators credentials were disclosed to the focus groups participants, and personal opinions were not expressed to minimise the risk of bias.

#### 2.4.2. Relationship with Participants

Participants were recruited via word of mouth, purposive and snowball sampling. To minimise relationship bias, we did not include any current patients of the research team.

### 2.5. Study Design

#### 2.5.1. Methodology

We employed a qualitative descriptive study design to capture the perceptions and experiences of people living with chronic lower-limb musculoskeletal injury and healthy physically active populations [57]. Qualitative descriptive methodology is suitable in healthcare research as it helps to focus research questions directly on participants’ experiences rather than through researcher interpretation or a theoretical lens [58].

Qualitative descriptive study designs are inherently simple yet flexible and allow researchers to perform descriptive research based on qualitative methodology [59]. Furthermore, some of the key features of qualitative descriptive research include small focus groups, purposive sampling, descriptive statistics, and thematic analysis [60], making this research design ideal when considering our objectives.

#### 2.5.2. Participant Selection

We recruited Australian participants over 18 years of age from two subgroups: (1) people living with chronic lower-limb musculoskeletal injury (>3 months of lower-limb pain or injury), and (2) healthy physically active populations. We performed focus groups until we achieved saturation, which was defined as no new concepts being introduced throughout an entire focus group.

### 2.6. Setting

Four focus groups were held in a series of locations, most convenient for the participants (*n* = 3 online, *n* = 1 in person), although primarily online using the MS Teams platform.

### 2.7. Data Collection

One week prior to commencement of the focus group, participants were sent a link to a Qualtrics survey for simple demographic details, as well as dietary preferences and allergens (which were factored into simple catering for the event). The following data were collected: age; gender; ethnicity; height, weight; level of education; annual household income; physical activity level; medical history including all chronic diseases, pains and injuries; dietary preferences and allergens (as focus groups were catered).

All focus groups were recorded using Microsoft Teams (version 23335.232.2637.4844). Following the conclusion of the first two focus groups, we employed a cognitive interviewing technique to ensure all questions were understandable and facilitated discussion that addressed the concept topic. Participants were specifically asked: (1) if there were any questions they did not understand and, (2) whether they felt the questions addressed the concept topics appropriately. However, no amendments were made to the questions.

The principal investigator (MCM) exported the data from Microsoft Teams (version 23335.232.2637.4844). Transcripts were automatically generated and cross-checked via a research assistant against the audio recording by another member of the research team (CH).

### 2.8. Analysis

A single researcher (CH) coded all data using QSR NVivo (version 12) under the guidance of a senior member of the research team (AT) who has substantial experience with NVivo for qualitative research analysis. The transcripts were first read line by line to identify significant statements that captured individual perspectives of the participants.

A thematic template was then created, following Braun and Clarke’s six stages of thematic analysis, and was used to create a coding structure [61]. This was developed from the initial skeleton code frame based upon the concept of interest and subsequently built upon during the coding phase. Codes were then grouped into subthemes, and four key themes were then derived from the data. Whilst only a single researcher coded the data (CH), the coding and themes were cross-checked by two members of the research team (AH: who identifies as a qualitative research expert; MCM: who conducted the focus groups and was able to confirm the themes generated from transcripts aligned with the experiences of the focus group).

## 3. Results

### 3.1. Participant Characteristics

We included 16 participants (44% women), with a mean (SD) age of 38.9 (14.8) years, height of 175.9 (9.6) cm and weight of 76.6 (19.8) kg. Participants had a variety of socioeconomic and occupational backgrounds with many different pain regions. Detailed participant characteristics are provided in Table 2.

### 3.2. Themes

Four primary themes were identified from the focus groups. These were (i) the impact of musculoskeletal injuries on health and quality of life, (ii) performance and injury-recovery as facilitators to using tDCS, (iii) barriers and facilitators to tDCS application and (iv) design and aesthetic factors for a tDCS device. Saturation was reported as achieved following the third focus group when no new themes emerged in the fourth focus group. Further, despite the recruitment of both pathological and healthy participants, other than ‘the impact of musculoskeletal injuries on health and quality of life,’ there were no differences observed between pathological and healthy participants in the other three themes.

#### 3.2.1. Impact of Musculoskeletal Injury on Health and Quality of Life

Participants shared their experiences of musculoskeletal injuries (Table 3). Participants had experienced many different injuries (with some having multiple injuries), such as anterior cruciate ligament rupture, ankle reconstructions, Achilles tendon rupture, shoulder subluxations, and chronic lower back pain from a variety of different sporting and non-sporting exposures. Three sub-themes were identified: (i) Musculoskeletal injuries have a negative impact on quality of life, (ii) Regardless of the region, injuries can be serious, and (iii) Trialling numerous management strategies.

#### 3.2.2. Performance and Injury-Recovery as Facilitators to Using tDCS

Participants discussed if they would be open to use the of tDCS (Table 4). Specifically, participants reported they would support the use of tDCS to facilitate recovery form injury or improve athletic performance, but some participants conveyed that brain stimulation was a concept is scary. The subthemes included (i) Facilitate recovery form injury, (ii) Improve athletic performance, and (iii) Brain stimulation can be scary.

#### 3.2.3. Barriers and Facilitators to tDCS Application

Participants discussed their perceived barriers and facilitators to the use of tDCS (Table 5). Many had a variety of different opinions, but these opinions appeared to be independent of their injury history. For example, a desired facilitator was to include clear instructions if they were planning to use tDCS. Conversely, participants identified barriers through concerns related to anything being wrapped around the head and whether the sensation of tightness could give rise to headaches. Participants identified the main barriers/facilitators to the use of tDCS were: (i) usability, (ii) discomfort, and (iii) hygiene.

#### 3.2.4. Design and Aesthetic Factors for a tDCS Device

Participants were clear that for tDCS to be adopted, the design and aesthetics of any device should be considered, which included (i) Aesthetics and (ii) Customisation (Table 6).

## 4. Discussion

This study identified key themes related to tDCS and improvements in disability and performance. Participants reported one of the primary reasons to consider an intervention like tDCS would be the substantial impact musculoskeletal injury has on health and quality of life. Participants reported that they would be willing to use interventions like tDCS if it was able to assist performance and speed up injury recovery. However, a number of barriers to the implementation of tDCS exist and participants felt an aesthetically pleasing tDCS would make implementation of tDCS more likely.

It is vital that the contextual needs of the population are considered in order for any intervention to be successfully implemented into clinical practice [62]. Participants in this qualitative study reported a number of barriers to the implementation of tDCS (e.g., fear related to the use of tDCS or discomfort of the device). Each of these barriers should then be considered and addressed if tDCS is to be used in routine practice.

The most prominent barrier we identified to tDCS use in this research study was a fear related to brain stimulation. Therefore, clinicians must address this fear to ensure implementation is successful. As tDCS has an excellent safety profile [63,64], patients need to be educated on this, which should reduce fear of non-invasive brain stimulation. Education related to tDCS should also include knowledge of the common adverse events, such as itching. For example, even people with stroke who have used tDCS and found it beneficial (90% reported that it was enjoyable and 100% reported that it was beneficial for their rehabilitation) reported having adverse events like itching and burning [65].

Discomfort and usability were identified as barriers to wanting to use tDCS. A vital element of tDCS application is that the sponges are sufficiently wet and make firm contact with the skin [66]. However, the contact/attachment/securement methods for electrodes in tDCS have been raised as a concern [67]. Therefore, firm contact is required, but this can be problematic when considering stimulation of the M1 region in the brain due to the location on top of the head and current devices typically using a headband or cap design. For the use of tDCS with exercise, such as resistance training in a gym, it is doubly important that a tDCS device is secure as it must remain in place even when sweating or moving around. However, if too secure, it could be an irritant and trigger headaches, which can be easily triggered by external compression [68], and lead to poor uptake.

One unexpected element of tDCS design that was raised during the focus groups was aesthetics. Participants reported that the likelihood they would use the device, particularly in public, would increase if it looked stylish. Furthermore, they reported that uptake might be better if people could tell what the device was by looking at it instead of just looking like they were wearing a strange device on their head. Thus, innovative approaches to device application that allow for a stylish design and clearly visualised purpose may be effective in successful implementation. This may include design elements that allow for cranial and non-cranial electrode placements (e.g., the design performed in Murphy et al., 2024 [45]), which are currently not possible with existing devices without substantial amendments—that would be beyond what is feasible in a clinical setting.

### 4.1. Future Research

Prior to designing an intervention, the experiences of the end-users (e.g., patients and/or athletes) should be established [69]. This study has established these experiences. Therefore, the next step is to design a tDCS intervention that addresses the barriers to implementation identified in this study.

### 4.2. Limitations

This qualitative study includes people from Perth, Australia, and the results may not be generalisable to other regions. However, we included participants of diverse ages, genders, occupations, education levels, athletic experiences and socioeconomic status for as generalisable results as possible.

## 5. Conclusions

Our qualitative descriptive study identified four themes relevant to the successful implementation of tDCS into rehabilitative and performance practice. A desire to facilitate injury recovery or performance enhancement will drive people to use tDCS. However, barriers such as a fear of brain stimulation, poor usability, discomfort, and unhygienic tDCS practices will reduce the likelihood of engaging with tDCS for rehabilitative or performance gains. Therefore, to increase the likelihood of successful tDCS implementation, these barriers should be addressed.

## Figures and Tables

**Table 1 neurosci-06-00072-t001:** Personal Characteristics of Interviewers.

Interviewer	Credentials	Occupation	Gender	Experience and Training
Myles Murphy	PT, PhD	University Academic and Physiotherapist	Man	Has conducted several qualitative research studies, including supervising an honours student performing qualitative research. Is an experienced sport physiotherapist having worked in both elite sport and private clinical practice.

Legend: PT = Physiotherapist, PhD = Doctor of Philosophy.

**Table 2 neurosci-06-00072-t002:** Participants characteristics.

Participant Characteristic		Count
Highest level of education	Less than high school degree	1
	High school degree	0
	Bachelor’s degree	13
	Master’s degree	1
	Doctoral degree	0
	Professional degree (JD, MD)	1
Occupation	Academic	1
	Cleaner	1
	Electrical Engineer	1
	Graphic Designer	1
	Medical Doctor	1
	Police Officer	1
	Physiotherapist	4
	Student	2
	Taxi driver	1
	Teacher	2
	Retired	1
Employment Status	Student	2
	Working part-time	3
	Working full-time	10
	Retired	1
Household Income	Between AUD $30,000–$49,999	1
	Between AUD $49,999–$79,999	0
	Between AUD $80,000–$99,999	6
	Between AUD $100,000–$149,999	2
	Between AUD $150,000–$199,999	5
	More than AUD $200,000	2
Medical Conditions	Depression	1
	Hypertension	1
Regions of pain and injury in the past 12 months	Ankles and Feet	2
	Knees	5
	Hips/Thighs	2
	Lower Back	7
	Thoracic Spine	0
	Neck	3
	Shoulders	5
	Elbows	1
	Wrists and Hands	2
Moderate to vigorous physical activity at least three time weekly	No	3
	Yes	13

**Table 3 neurosci-06-00072-t003:** Impact of musculoskeletal injury on health and quality of life.

Sub-Theme	Supporting Evidence
Musculoskeletal injuries have a negative impact on quality of life	“It changes the type of exercise that I can do. Like I actually I thought that my back hurting a little bit was normal for a long time.” “I am a fear avoidant of running. I used to do a lot of marathons, but probably a little bit like to stick to about 10 km” “It has impacted things, like my ability to just like go out to hike, like all those things” “It’s been quite a long journey since I haven’t been able to run until like the last two or three months. I run a lot and cycle as well, so thankfully I was able to get into cycling a little bit earlier. It’s part of the recovery process.” “I’ve been doing Brazilian Ju Jit Su for a long time as well, and running, you know, I love running. Yeah. So I tried to get into other things”
Regardless of the region, injuries can be serious	“I’ve had a bunch of ankle issues, like rolled ankles. So now my ankles are super fat. And that’s just what they look like. I had surgery on one the other day and they took 13 pieces of little chips of bones and stuff out of it.” “I tore my right Achilles tendon in 2016” “I have fractured my left ankle in the past so now I have osteoarthritis, and I have chronic tendonitis” “I tore my right ACL in 2018, and after a year I tore my left ACL” “In my head. I’m like, I’ve never really been injured, but I’ve had four or five years of chronic back pain.” “I have always had a like weaker back as well. It’s always brought pain down the left side of my body.”
Trialling numerous management strategies	“I had surgery for osteitis pubis when I was 18.” “I had a hamstring avulsion water skiing where they had to like screw the hip back on over the tendon, then stitch the muscle back together.” “One (injury) is a shoulder where I had a cyst removed surgically” “After surgery (I performed) physio recommendations and follow those exercise routines”

**Table 4 neurosci-06-00072-t004:** Performance and injury-recovery as facilitators to using tDCS.

Sub-Theme	Supporting Evidence
Facilitate recovery form injury	“I think I’d use it (tDCS) if I was at a doctors or a rehab facility.” “I wouldn’t do it (tDCS) from a television advertisement, but I’d do it from a doctor’s recommendations.” “If it (tDCS) meant reducing the pain (I would use it).” “I’d do it (tDCS) on a professional’s recommendation.” “You’d be really wanting to see someone for an assessment and have them advise whether or not they think it’s a good idea for you to be using it (tDCS).” “I would just go to the physio or doctor and if they recommended it, I would do it.”
Improve athletic performance	“Honestly, as long as it improves my performance, I don’t really care that much because I know that it works. And I know that it will really help.” “But if you found it was working, I think then going to a gym and people see it’s working, they don’t care what you look like.” “Most athletes don’t really care. But some athletes, I would imagine, would be like if you’re going to give me a benefit I will wear it. Yeah. Like the athletes that I would deal with, they don’t care even about pain.”
Brain stimulation can be scary	“Brain stimulation is, I think it’s scary for people.” “The horror that comes to mind is shock therapy where they used to for mental patients give them shock therapy and that had nothing but bad news and that was electrical currents.” “I think you have to gently educate people rather than just, because anything to do with brain stimulation is scary.” “Oh, well, we’re going to do some brain, you know, stimulation. You’re thinking, jeez, hang on a second.” “I think straight off, you know, you have a broken ankle. We’re going to do your brain. That’s a little scary for me anyway.” “(Can) you get people who are obsessive, and can you overcook yourself with this thing (tDCS)?”

**Table 5 neurosci-06-00072-t005:** Barriers and facilitators to tDCS application.

Sub-Theme	Supporting Evidence
Usability	“How easy/difficult will this be to replace parts/fix?” “I could do 20 min of wearing it around the house before the gym and then. Yeah, leave it at home. And do. Yeah. Drive into the gym.” “You’d need good instructions or something to be able to set it up yourself.” “Knowing like how much voltage I’m meant to be getting through, how much is too much or too little stuff like that.” “Is it easy to change the little sponges?”
Discomfort	“If it is really firm against your skull, it’s probably going to cause you a headache when you’re running.” “How does it move when I’m like, you know, running, how does it shift around perspiration.” “I imagine it will feel like headphones, even though it might not. But I don’t like wearing headphones because of the weight of the headphones.” “The thing that bothers me the most is how comfortable it will be, especially if I’m running.” “Then (what) if you’re sweaty and then it slides down.”
Hygiene	“So they (sponges) would have to be replaced disposable or something, wouldn’t they?” “If you used your own sponges, but if the clinic wiped it all over like they do with other sanitising, that would be fine.” “Couldn’t you have linings?” “As long as they (tDCS) have cleaning.” “Do they have a procedure for cleaning?”

**Table 6 neurosci-06-00072-t006:** Desire for a tDCS device that is consumer-friendly and aesthetically pleasing.

Sub-Theme	Supporting Evidence
Aesthetics	“People wearing it outside of being in like an actual physio gym or something like you want to be like, streamlined not bulky, something like hanging back more or like didn’t come down to the brow.” “I’ll be way less embarrassed to wear whatever it is if I think that anyone who looks at it can tell what it is.” “To make it more appealing to use in a gym could you integrate headphones in? Or just make them look like headphones.” “Incorporate into some sort of like hats or something, it doesn’t look too bad” “People wear hats when they play golf and everything, so it looks like if it’s more streamlined it’d like slot in a pretty easy.”
Customisation	“If it just had the visor, you could just adjust it” “(In terms of adjustable fit) I want the option It’d be nice to have the option.”

## Data Availability

De-identified data for this research project is available from the corresponding author upon reasonable request.

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
