# Peer review of "The Barriers and Facilitators to the Application of Non-Invasive Brain Stimulation for Injury Rehabilitation and Performance Enhancement: A Qualitative Study"

_neurosci, 2025, doi:10.3390/neurosci6030072_

Round 1
Reviewer 1 Report
Comments and Suggestions for Authors
The study presents limited utility, especially considering that the tDCS methodology intended to reduce cortical motor inhibition in chronic pain conditions has modest clinical evidence, particularly for lower limb pathologies. The results of studies on the effectiveness of tDCS in reducing pain and disability in LBP are conflicting: some studies have reported pain reduction and functional improvement with active tDCS compared to sham stimulation, while others have found no significant differences. This study investigates a delicate but relevant topic: the perceptions and experiences of tDCS users. The proposed qualitative methodology is appropriate; however, the sample size of 16 participants with at least 2 groups is modest, particularly given the heterogeneity of the participants (pathological vs. healthy), and raises doubts about the possibility of achieving adequate thematic saturation. Although the study confirms the relevance of known barriers and facilitators within this specific context, its primary contribution seems to be this contextualization rather than the evidence of new insights. The premise regarding the efficacy of tDCS is an oversimplification of the current evidence base and is still quite uncertain. The highlighted results do not concern the effectiveness of the treatment but rather the identification of the main themes that need to be considered in the application of tDCS for chronic pain problems, particularly in the lower limbs. The introduction could be partially expanded to describe the activity of tDCS at the cortical level, particularly regarding the modulation of chronic pain.
Author Response
REVIEWER ONE
- The study presents limited utility, especially considering that the tDCS methodology intended to reduce cortical motor inhibition in chronic pain conditions has modest clinical evidence, particularly for lower limb pathologies. The results of studies on the effectiveness of tDCS in reducing pain and disability in LBP are conflicting: some studies have reported pain reduction and functional improvement with active tDCS compared to sham stimulation, while others have found no significant differences. This study investigates a delicate but relevant topic: the perceptions and experiences of tDCS users.
- Thank you for this comment. We recognise there have been inconsistent results of tDCS in prior studies (particularly in lower-back pain). To address this we have now included several recent systematic review and meta-analyses as citations (e.g., PMID: 39664709) that demonstrate an overall positive effect of tDCS on pain in lower limb conditions, such as knee osteoarthritis. We have also highlighted that elements, such as the number of treatment sessions, that may influence outcomes. (Page 2)
“Non-invasive brain stimulation has been shown to directly alter the amount of motor cortex inhibition [45, 46], and the efficacy of non-invasive brain stimulation appears to be associated with the number of treatment sessions [47].
Clinical trials have demonstrated the benefits of non-invasive brain stimulation for musculoskeletal conditions, such as knee osteoarthritis and anterior cruciate ligament reconstruction [40, 45, 48-50].”
- The proposed qualitative methodology is appropriate; however, the sample size of 16 participants with at least 2 groups is modest, particularly given the heterogeneity of the participants (pathological vs. healthy), and raises doubts about the possibility of achieving adequate thematic saturation.
- We agree this could be a problem. However, we achieved saturation after the third focus group and other than the first theme, there were no observable differences between groups. Thus, we have included this in the text. (Page 7)
“Saturation was reported as achieved following the third focus group when no new themes emerged in the fourth focus group. Further, despite the recruitment of both pathological and healthy participants, other than ‘the impact of musculoskeletal injuries on health and quality of life,’ there were no differences observed between pathological and healthy participants in the other three themes.”
- Although the study confirms the relevance of known barriers and facilitators within this specific context, its primary contribution seems to be this contextualization rather than the evidence of new insights.
- We agree that many of our identified clinical barriers to non-invasive brain stimulation are known to clinicians. However, there has been no research to date that has confirmed this. Further, this study presents the facilitators that can be used to overcome the barriers to tDCS use, which is what makes this scientific contribution novel.
- The premise regarding the efficacy of tDCS is an oversimplification of the current evidence base and is still quite uncertain.
- Thank you for this comment, as per our response to comment one, we have amended this to provide more recent meta-analyses related to the efficacy of the tDCS in lower-limb conditions and that outcomes may be influenced by the number of treatment sessions.
“Clinical trials have demonstrated the benefits of non-invasive brain stimulation for musculoskeletal conditions, such as knee osteoarthritis and anterior cruciate ligament reconstruction [40, 45, 48-50].”
- The highlighted results do not concern the effectiveness of the treatment but rather the identification of the main themes that need to be considered in the application of tDCS for chronic pain problems, particularly in the lower limbs.
- We entirely agree, this was the aim of our paper.
- The introduction could be partially expanded to describe the activity of tDCS at the cortical level, particularly regarding the modulation of chronic pain.
- We agree this should be acknowledged. We have now included this within the introduction to discuss the proposed role of tDCS at improving the conditioned pain modulation effect. (Page 2)
“One of the mechanisms by which non-invasive brain stimulation is proposed to work in people with chronic pain is restoring impaired pain modulatory mechanisms, such as conditioned pain modulation [48, 51]. Despite the evidence for efficacy in reducing pain, there has been minimal uptake of non-invasive brain stimulation interventions, such as transcranial direct current stimulation (tDCS), for chronic musculoskeletal conditions in clinical practice [52].”
Reviewer 2 Report
Comments and Suggestions for Authors
Trying to better understand barriers to the use of tDCS is helpful in determining how to improve acceptability of the intervention. I have concerns however about the methods.
- results of injured and noninjured participants were combined. were the demographics similar between the two groups? also there were 4 physiotherapists in the groups which might bias results.
- all of the coding was done by a single individual. there was no attempt at assessing reliability of coding.
- I have not seen a table describing the qualification of the interviews in a paper like this before. could be briefly provided in text, no table needed
Author Response
REVIEWER TWO
- Trying to better understand barriers to the use of tDCS is helpful in determining how to improve acceptability of the intervention.
- Thank you, that was the aim of this study and we hope this can help clinicians aiming to use this technology.
- I have concerns however about the methods: Results of injured and noninjured participants were combined. Were the demographics similar between the two groups? also there were 4 physiotherapists in the groups which might bias results.
- Thank you for this comment. We have now more formally acknowledged the diversity of the sample (which we view as a strength), whilst reporting that other than the first theme, which was only commented on by pathological participants, there was no difference between the groups. (Page 7)
“Saturation was reported as achieved following the third focus group when no new themes emerged in the fourth focus group. Further, despite the recruitment of both pathological and healthy participants, other than ‘the impact of musculoskeletal injuries on health and quality of life,’ there were no differences observed between pathological and healthy participants in the other three themes.”
- All of the coding was done by a single individual. There was no attempt at assessing reliability of coding.
- Thank you for this comment, and yes we did have only one person perform the coding. However, we did perform cross-checking and have now made this amendment. (Page 4)
“Codes were then grouped into subthemes, and four key themes were then derived from the data. Whilst only a single researcher coded the data (CH), the coding and themes were cross-checked by two members of the research team (AH: who identifies as a qualitative research expert; MCM: who conducted the focus groups and was able to confirm the themes generated from transcripts aligned with the experiences of the focus group).”
- I have not seen a table describing the qualification of the interviews in a paper like this before. could be briefly provided in text, no table needed.
- Thank you for this comment. We have discussed this as a research team and have opted to keep the current format, which aligns with previous qualitative research we have conducted. We have this is acceptable as an author formatting preference.
Reviewer 3 Report
Comments and Suggestions for Authors
The authors’ qualitative descriptive study identified four themes relevant to the successful implementation of transcranial direct current stimulation (DCS) into rehabilitative and performance practice. A desire to facilitate injury recovery or performance enhancement will drive people to use tDCS. However, barriers such as a fear of brain stimulation, poor usability, discomfort, and unhygienic tDCS practices will reduce the likelihood of engaging with tDCS for rehabilitative or performance gains.
The authors concluded that to increase the likelihood of successful tDCS implementation, these barriers should be addressed.
In the Abstract, the Results Section is actually a part of the extension of the Methods Section, and most of the Discussion Section, except the last sentence, is actually results. The authors are suggested to rewrite this part, and perhaps to elaborate more on the Discussion regarding the pros and cons of tDCS.
Overall, this is a well-written and well-delineated manuscript. It would be encouraging that authors could come up with ideas and approaches to facilitate the use of tDCS to improve musculoskeletal conditions.
The inclusion of the authors’ titles in the authorship is not a common practice and could lead to confusion in citation. Please remove the titles.
Author Response
REVIEWER THREE
- The authors’ qualitative descriptive study identified four themes relevant to the successful implementation of transcranial direct current stimulation (DCS) into rehabilitative and performance practice. A desire to facilitate injury recovery or performance enhancement will drive people to use tDCS. However, barriers such as a fear of brain stimulation, poor usability, discomfort, and unhygienic tDCS practices will reduce the likelihood of engaging with tDCS for rehabilitative or performance gains. The authors concluded that to increase the likelihood of successful tDCS implementation, these barriers should be addressed.
- Thank you this is an excellent summary.
- In the Abstract, the Results Section is actually a part of the extension of the Methods Section, and most of the Discussion Section, except the last sentence, is actually results. The authors are suggested to rewrite this part, and perhaps to elaborate more on the Discussion regarding the pros and cons of tDCS.
- We agree the discussion was a repeat of the results and we have amended the discussion to state what is needed to improve the implementation of tDCS.
“Our qualitative descriptive study identified four themes relevant to the successful implementation of tDCS into rehabilitative and performance practice. To increase the likelihood of successful tDCS implementation, these barriers should be addressed and facilitators promoted. This should include innovative approaches to device application and structure that allow for a stylish, user-friendly design.”
- Overall, this is a well-written and well-delineated manuscript. It would be encouraging that authors could come up with ideas and approaches to facilitate the use of tDCS to improve musculoskeletal conditions.
- Thank you for this comment – we have made some amendments to the discussion to help address this point.
“This may include design elements that allow for cranial and non-cranial electrode placements (e.g., the design performed in Murphy et al. 2024 [45]), which are currently not possible with existing devices without substantial amendments – that would be beyond what is feasible in a clinical setting.”
- The inclusion of the authors’ titles in the authorship is not a common practice and could lead to confusion in citation. Please remove the titles.’
- Thank you for this comment we have made this amendment as suggested. (Page 1)